# How Naive Is Contentful Moral Perception?

**Preston J. Werner**

Department of Philosophy, Hebrew University of Jerusalem, Jerusalem 9190401, Israel;
preston.werner@mail.huji.ac.il

**Abstract:** According to contentful moral perception (CMP), moral properties can be perceived in the same sense as tables, tigers, and tomatoes. Recently, Heather Logue (2012) has distinguished between two potential ways of perceiving a property. A Kantian Property (KP) in perception is one in which a perceiver's access involves a detection of the property via a representational vehicle. A Berkeleyan Property (BP) in perception is one in which a perceiver's access to the property involves that property as partly constitutive of the experience itself. In this paper, I set aside generalized arguments in favor of one view or another, and instead ask whether proponents of CMP have reasons to understand moral perception as Kantian or Berkeleyan. I explore three possible explanatory differences—(a) explaining the intrinsic motivating force of moral perceptions, (b) providing a metasemantics for moral properties, and (c) providing an epistemology of the normative authority of moral properties.

**Keywords:** moral perception; representationalism; naive realism; normative authority

A plurality of contemporary analytic philosophers accept representationalism about perceptual experience[1]. However, representationalism has a significant and growing number of detractors, many of which endorse one of a variety of versions of naive realism, or presentationalism. The dispute between these two views has largely taken place at the general level of the metaphysics of perceptual experience, as opposed to more localized disputes about the nature of some particular property (re)presented in experience. However, Heather Logue [1,2], in recent work, has raised and defended the possibility of a more complicated view. On this view, like traditional naive realism, perceptual experience is constituted by a relation between subject and world, but the contribution by each relatum in a given experience can vary greatly between properties. In other words, some aspects of perceptual experience involve direct acquaintance with worldly features, while others involve indirect access, since it is the subject's cognitive states that more wholly determine the phenomenology[2]. Following Logue's [1] terminology, we can call the latter properties *Kantian Properties* and the former properties *Berkeleyan Properties.*

Moral perception, as I'll use it here, is the view that perceptual experience sometimes contains moral properties as part of its content. When such a view is precisified, it is almost always precisified in representationalist language [3–7][3]. But none of the central arguments for the view require a commitment to representationalism, as opposed to naive realism, in any of their variety of forms. Presumably (and this is just an educated guess), proponents of moral perception are generally happy to defer to whatever the philosophers of mind tell us about the fundamental metaphysical nature of perceptual experience, and thus speak in representationalist-friendly terms, since it is the closest thing to a 'received' view there is in the literature. And moral perception is on shaky enough philosophical grounds that it would be unwise to use it to leverage support for one of the two general theories of the metaphysics of perceptual experience. So until now, if the moral perceptualist is asked "Are moral properties presented or represented in perceptual experience?", the sensible answer for her to give is simply "That's above my paygrade—whatever is true of the other properties is also true of moral properties".

But now suppose that we accept a distinction like that which Logue proposes. This would make the answer just given unacceptable. On Logue's view, we have (at least!) two buckets of properties in perceptual experience: Kantian Properties and Berkeleyan Properties[4]. It's an open question which bucket the moral properties should go in, assuming moral perceptualism is true. And, as we will see, given that the answer to this question will depend both on the nature of the moral properties as well as the phenomenal signature of those properties in perceptual experience, this question appears to fall directly under the purview of the moral perceptualist interested in developing a complete positive view.

The aim of this paper is to explore these questions. I don't intend to provide any novel arguments in favor of representationalism or naive realism—full stop[5]. Rather, I want to explore, within Logue's framework, whether we have reasons to posit that moral properties in experience are Kantian or Berkeleyan. Following Logue, I will adopt the claim that naive realism about a property *F* will provide a distinctive, non-propositional kind of epistemic relationship to *F* that the representationalist cannot easily explain. Then, I will explore three arguments that ensuring this kind of epistemic relationship can be met is uniquely and especially important when it comes to moral properties. If any of these arguments work, this would mean that there are powerful (albeit defeasible) reasons for the moral perceptualist to argue that our access to moral properties in perception is *presentational*. However, assessing the three arguments, my conclusions are mixed—in one case, the argument does not seem to provide any reason to favor a presentational account of moral properties. In the other two cases, the success of the argument depends on other metaethical commitments which are contentious.

The plan of the paper is as follows. In Sections 1.1 and 1.2, I provide a brief overview of representationalism and naive realism as theories about the metaphysics of perceptual experience. In Section 1.3, I discuss Logue's theory of perceptual experience, the *Extended Theory* (as she calls it), which allows for a bifurcation between Kantian and Berkeleyan Property perception. Logue's theory opens up more fine-grained questions about the relationships between different aspects of perceptual experiences and the world. In Section 2, I lay out the commitments of moral perceptualism as I will understand it for the purposes of this paper. I do this with a special eye toward formulating the claims in a way that is neutral between representationalism and naive realism. The previous sections provide the necessary background for Sections 3.1–3.3, where I directly address the question of whether moral perceptualists should think of moral properties in perceptual experience as Berkeleyan or Kantian. I examine several considerations—epistemic, motivational, and metasemantic—that may tell in favor of conceiving moral properties as Berkeleyan (if anything is). I conclude that these considerations *do* defeasibly tell in favor of a Berkeleyan interpretation, at least for a certain kind of normative realist. I then sum up.

## 1. The Metaphysics of Perceptual Experience

As I write this, I am sitting at my desk. A moment ago, wanting to look over the notes I made for this section, I glanced over at my small notebook, taking in the nice shades of orange and lavender and the series of blue triangles which make up its cover design. In short, I had a visual experience of the small colorful notebook. The question that the representationalist and the naive realist are both trying to answer is this: What is the metaphysical nature of this visual experience? Put another way, what does it consist of?

An answer to this question should arguably explain, or explain away, three seemingly distinctive characteristics of perceptual experience [8][6]:

> *Phenomenological Characteristic:* Perceptual experience, at least paradigmatically, exhibits a certain kind of phenomenal character—there is a 'what it's like' to perceive objects and properties (such as the qualia of the various colors on the cover of my notebook).

*Epistemological Characteristic:* Perceptual experience, at least paradigmatically, plays an essential role in justifying beliefs about our immediate surroundings (such as justifying my belief that my notebook is on the table).

*Behavioral Characteristic:* Perceptual experience, at least paradigmatically, plays an essential role in facilitating action (such as reaching out and grabbing my notebook).

With these three characteristics in mind, we can turn to a brief sketch of both representationalism and naive realism, and then how each of these theories aims to explain each of these characteristics. (A reader familiar with representationalism and naive realism can skip ahead to Section 1.3).

### 1.1. Representationalism

Consider the following English sentence, uttered in the context of us sitting around my desk:

(N) "There is a notebook on the desk".

This sentence, uttered in this context[7], can express something true or false. It is true just in case *there is a notebook on the desk.* But (N) doesn't actually *contain* a notebook, a desk, or any kind of 'on-top-of' relation. It merely represents those features, such that the statement has correctness conditions which are met iff the world is a certain way.

Representationalism about perceptual experience says that, like sentences (or beliefs, or maps), perceptual experiences are representations. Just as (N) does not actually contain any notebooks or desks as its parts, a perceptual experience of the notebook on the table is not constituted out of any worldly objects like the notebook or the table that I am seeing. Just as with (N), the perceptual experience is constituted out of representational vehicles— in the case of the experience, not words or syntactic structure, but rather phenomenology (and its parts) or perhaps some subdoxastic states which underlie it.

Representationalism has much going for it [8][8]—which is why it is perhaps the most popular theory of the metaphysics of perceptual experience, in one form or another [9][9]. However, philosophers have raised worries about both its ability to explain the phenomenological characteristics of perception as well as its epistemological characteristics [10,11]. My aim here is not to adjudicate these objections and varieties of responses to them; what matters for the present purposes is that representationalism has its weaknesses. These weaknesses, as well as recent developments [12][10], have motivated many philosophers of perception toward a return to naive realism.

### 1.2. Naive Realism

Notice that for representationalists, our access to the world is, in an important sense, *indirect*. The word "notebook" can represent notebooks even though it does not resemble them in any way. And certainly, the word "notebook" is not—even partly—composed of notebooks. Similarly, there need be no intrinsic resemblance between, say, qualitative red and the worldly property of *redness,* or the qualitative nature of my notebook experience and the worldly notebook itself. Qualitative redness and qualitative notebook-ness do not need to contain *redness* or *being-a-notebook* as any of their parts. This isn't to say that representational vehicles *couldn't* resemble their referents at least in some respects, but it is to say that such resemblance is inessential to the accuracy conditions of the representations themselves. The representationalist is denying two separate but related claims here:

*Acquaintance:* For S to have a perceptual experience *e* of *F*, *e* must provide S with *direct awareness* of *F*.

*Constitution:* For a perceptual experience *e* to be an experience of *F*, *e* must be partly constituted by *F*.

It's essential to naive realism, as I'll understand it here, to endorse *Constitution.* Naive realists take veridical perceptual experiences to be constituted by a relation between per-



ceiver and some chunk of the world such that the chunk of the world is an essential relata in a given veridical perceptual experience. A given veridical perceptual experience, then, doesn't just *represent* some external state of affairs, but *presents* it, in the sense that it couldn't exist without the presented object as one of its constituents.

To see why naive realists also endorse *Acquaintance*, consider a non-perceptual case where a partial constitution relation holds: There is a set of neurons, *N*, which partly constitute my cognitive system, *S*. Nonetheless, this partial constitution relation doesn't put me in any special position to form beliefs about the nature or structure of *N*—in fact, I'd probably be in a worse position than a neuroscientist who was looking at that patch of neurons using an fMRI. Why, then, would the fact that a perceptual experience is partly constituted by, say, a notebook, put one in a position to form justified beliefs about that notebook?

The answer that naive realists standardly appeal to is the notion of *Acquaintance*. For naive realists, veridical perceptual experiences provide direct, unmediated access to the world in such a way that they acquaint us with some chunk of the world. Like sense data theorists such as Russell [13,14][11], naive realists believe that perceptual experience involves the direct awareness, by the perceiver, of the perceived objects/properties. The difference is that, while sense data theorists posit sense data as the objects/properties of experience, the naive realist extends direct, unmediated awareness to the mind-independent world[12] [15,16]. Unlike any ordinary part/whole relation, naive realists can ensure the justification of beliefs about external world objects and properties by positing our direct awareness of them, which can underlie a powerful epistemic relationship between perceiver and perceived.

Furthermore, *Acquaintance* can be independently motivated. We can see how by considering some remarks from John Campbell concerning Putnam's (in)famous argument that brain-in-vat skepticism is incoherent. Putnam's argument, recall, is that the brain-in-vat skepticism argument cannot work, since if we were envatted, given a causal theory of reference, our terms would have different referents which were fixed by whatever brain stimulation causally triggered those concepts. For example, CHAIR, in such a situation, would refer to certain brain stimulations which caused the CHAIR concept to be tokened, so in such a scenario it *would* be true that "there are chairs". This argument has struck many philosophers as unsatisfying, but it hasn't always been easy to articulate why. However, as Campbell hypothesizes:

> The reason Putnam's Proof is intuitively so unsatisfactory is that we ordinarily take experience to provide us with knowledge of far more than merely the functional structure of the medium-sized world. We take ourselves to have knowledge of the categorical objects and properties around us. We ordinarily think we know what the world is like. If the world is that way, it is not a bit like a vat. [11], p. 151.

Campbell's claim here is that our knowledge of the external world around us appears to be more than merely a knowledge of some structural relationship between things that cause such and such experiences, whatever those things might be. When we believe that *roses are red*, we take the truth conditions of that belief to be something more than just that we have a concept, ROSE, and a concept RED, such that the tokening of those concepts is generally, standardly, or teleologically related to or caused by some things in the world that happen to make the belief true. Instead, we think that we have some further intrinsic (in Campbell's word "categorical") understanding of what *roses* and *redness* are like. And this intrinsic knowledge requires *Acquaintance* with worldly properties and objects. Only naive realism is strong enough to ensure this kind of relation—at least according to many naive realists.

### 1.3. Kantian and Berkeleyan Properties

So far, I have briefly summarized representationalism and naive realism. I did this, not with an eye toward making progress on resolving this dispute, but to clear the groundwork for introducing a distinction from Heather Logue (2012). Logue is a naive realist, but what we might call a *moderate* naive realist. Unlike traditional naive realism, moderate naive realism allows that perceptual experiences' character can fail to provide *Acquaintance*, since in some cases, the qualitative character of some object or property presentation will be determined more by the inner workings of the *perceiver* and less by the intrinsic nature of the *perceived*. What this means is that, according to moderate naive realism, the kind of epistemic benefit flagged by Campbell and other naive realists will actually only be present for some of the perceived objects and properties, depending on which side of the perceptual relation is doing more of the work in grounding that experience's phenomenal character.

Following Logue, then, we can distinguish between two types of properties in perceptual experience (and of course each given experience can contain a mixture of both):

A *Kantian Property* is a property such that the phenomenal character of veridical experiences of them are mostly determined by features of the subject. [1], p. 214.

A *Berkeleyan Property* is a property "such that the phenomenal character of veridical experiences of it is mostly determined by the fact that the subject perceives an instance of that property (that is, features of the subject play a relatively minimal role in determining phenomenal character)". [1], pp. 226–227.

Kantian Properties are so called in reference to the fact that, on Kant's view, there is a veil between our experience and the world. Berkeley rejected this veil, thus Berkeleyan Properties. Believing a property is Berkeleyan is *not* to be committed to any, even limited form, of idealism. As Logue—in line with the arguments given above—explains, this distinction between Kantian Properties (KPs) and Berkeleyan Properties (BPs) allows us to draw a line between two kinds of roles that perceptual experiences can play. On the one hand, a veridical experience can "put its subject in a position to know that certain properties are instantiated by things in her environment". On the other hand, a veridical experience may also "put its subject in a position to know what the things that instantiate those properties are like *independently of experience*". [1], p. 228. While both KPs and BPs can play the former role, only BPs could play the latter role; arguably this is precisely because BP-perceptions involve *Acquaintance*, while KP-perceptions do not.

Let me make two notes about the KP/BP distinction before turning to the central question of this paper (whether we should think of moral perception as KP or BP perception). First, the distinction—as has been noted—is quite similar to David Chalmers' distinction between edenic and non-edenic perception [17]. My inclination is to treat these distinctions the same for the purposes of this paper—if they are not variations of the same distinction, they are at least similar enough for my (largely exploratory) purposes here[13]. Second, even the formal definitions of KPs and BPs given above are imprecise: Whether something is a KP or a BP in some particular experience is a matter of degree (as Logue herself says). The amount to which some given experience of some given property is a KP or a BP depends on how much features of the subject contribute to that experience's phenomenal character. But there is no (and perhaps couldn't in principle be a) metric with some precise threshold to determine whether some property is a KP or a BP. And precisely because there is no precise threshold, I am satisfied enough—and I hope the reader will indulge me—with the rough distinction as it stands.

## 2. Contentful Moral Perception and Kantian/Berkeleyan Properties

Moral perceptualism is the view that perception and perceptual experience can be attuned to moral features in our environment. Moral perception comes in a variety of types; in what follows, I will begin by focusing on what I'll call *Contentful Moral Perception*, or CMP:

*Contentful Moral Perception* (CMP): An agent can represent moral properties as part of the content of her perceptual experience (along with shape, color, pitch, etc.)[14]. [18].

CMP is a quite ambitious thesis, and as such, is a matter of controversy on several grounds[15] [19,20]. But one thing which has not been explicitly noted is CMP's commitment to representationalism about perceptual experience. I think this is a mistake: As far as I know, *no* argument in favor of anything like CMP relies on representationalist-specific claims[16] [21–24]. Most likely, what has happened is that proponents of moral perception, being as they are largely focused on the 'moral' part of moral perception, simply have defined their thesis in terms of representation because it is the most popular theory of the metaphysics of perceptual experience, and it allows for a straightforward statement of the moral aspect of the kind of view they are defending.

Thankfully, this mistake is easily remedied, and without (as far as I can tell) undermining the previous literature on CMP. I propose a more ecumenical version of CMP:

CMP*: Moral properties, in human agents, are (re)presented as a content/constituent of their perceptual experience (along with shape, color, pitch, etc.).

CMP* is compatible with the arguments given in favor of CMP. Furthermore, it is compatible with all of the purported epistemic advantages raised by proponents of CMP, such as guiding deliberation by honing in on morally relevant features of one's situation and non-inferentially justifying moral beliefs[17] [4,25–30]. CMP* is neutral between representationalism, naive realism, or a hybrid view. That, in itself, is an advantage at demarcating what the unique and contentious philosophical claim behind moral perception is. But it also allows us to ask questions that the moral perceptualist may not have previously been able to ask: What are the advantages and disadvantages of thinking of moral properties as Berkeleyan? What are the arguments in favor of or against one or another view with respect to the moral properties in particular?

## 3. In Favor of Moral Perception as Berkeleyan

I turn now to explore three considerations that could be raised in favor of understanding the perception of moral properties as Berkeleyan. According to these three arguments, Berkeleyan moral perception would help to (i) explain issues in moral motivation, (ii) assist in a metasemantic story for moral properties, and (iii) explain epistemic access to the relevant *normativity* of moral properties.

I want to be clear about my aims here. I am considering conditional claims about some of the positive features that could result from a conception of moral properties as BPs. Even if these claims were all true, it would not prove that moral properties are BPs. For one thing, in each of these cases, there are alternative explanations of each of these data points. For another, there are, as mentioned above, completely general arguments against considering *any* perceptual properties as BPs—for example, perhaps traditional representationalism is true. If representationalism is true, establishing the conditionals discussed below would be a Pyrrhic victory. But given that the possibility of BPs remains a serious contender, even establishing the conditionals is worthwhile.

### 3.1. Moral Motivation

One feature of moral perception, as with moral judgment, is that the recognition of something as morally right/wrong/good/bad has motivational force. To hone in on a moral perceptual case, consider Harman's classic case of seeing some children pouring gasoline on a cat and lighting it on fire [31] Intuitively, the decent person's immediate and non-deliberative reaction to such a scene is to try to step in to save the cat. As Jean Moritz Müller says, "It is a familiar part of ordinary experience that our perceptible surroundings demand action from us [32], p. 3572." Even setting aside the phenomenological considerations, Müller also points out that "An important explanatory application of the idea that we experience prescriptive affordances concerns the normativity of non-deliberative action. This idea seems to make good sense of the fact that many of the actions we perform

are apt for normative assessment despite being unreflective" [32], p. 3573. Call this the 'perception-to-action' intuition.

Assuming these considerations are correct, how best can we explain this spontaneous motivational profile of moral perception? Compare two families of theories about motivation:

> *Humeanism:* "Necessarily, whenever an agent engages in some motivated action, φ, the complete explanation of her action must cite one or more of her desires as the ultimate source(s) of motivation to φ". [33].

> *Anti-Humeanism:* Sometimes, when an agent engages in some motivated action, φ, the complete explanation of her action will not appeal to any of her desires as the ultimate source(s) of motivation to φ.

In order to explain Harman's case compatible with the moral perceptualist explanation, it looks like the Humean would have to argue that a desire (or some other conative state) is able to rapidly influence the perceptual experience to result in a spontaneous motivation. In other words, the motivational content would need to somehow become incorporated into the perceptual experience itself (if we are to take the directness of the perception-to-action intuition at face value). But notice that this permeation would have to occur only *after* the recognition (in perceptual processing) of the presence of a moral property, lest the desire to be moral (or whatever, as the case may be) be irrelevant to any motivational upshot. The idea of background desires permeating experience in this way is not unprecedented or impossible, but it is more complex than the *Anti-Humean* explanation, at least once one is committed to spontaneously motivating moral perception[18].

However, the *Anti-Humean* explanation of motivation in moral perception also leaves us with a mystery: Why are moral properties, when perceived, able to intrinsically motivate? Understanding the moral properties in perception as BPs provides one plausible answer to this question. Kantian Perception of moral properties would provide us with propositional justification that some moral properties are present. But without some further explanation or background mental states, this wouldn't be enough to explain the motivational upshot of this knowledge. On the other hand, Berkeleyan Perception of moral properties would not just tell a perceiver that some moral properties are present, but may also provide access to their normative force. Acquaintance (in the sense that Berkeleyan Perception provides) with a moral property would pack in this normative component in a way that makes spontaneous non-deliberative motivational response rational[19].

One may think that some form of Motivational Internalism could provide an alternative explanation of the motivation (often) packed into moral perception. *Motivational Internalism* is related to *Anti-Humeanism.* But whereas Anti-Humeanism is a claim about the possibility for moral judgments to intrinsically motivate, Motivational Internalism says that there is a strong—perhaps even necessary—connection between moral judgments and motivation[20] [34]. If we extend Motivational Internalism to any moral representations (not just moral beliefs), we would have an explanation of the motivational nature of moral perception without requiring Berkeleyan Perception.

Certainly, a perception-inclusive version of Motivational Internalism would entail the motivating force of moral perception. But so far, this is just the illusion of progress. For now, we can ask what grounds Motivational Internalism—that is: *Why* do moral perceptions reliably motivate action? Perception-Inclusive Motivational Internalism says that they do, but it doesn't purport to answer why. Again, a natural answer to this question is to appeal to the status of moral properties in perception as Berkeleyan. One could alternatively claim that Kantian Perception of moral properties could also involve a second-order representation of the properties as action-guiding[21]. This is certainly possible, but unless the action-guidingness of moral properties is causally/teleologically related to the relevant phenomenology, it is unclear how it could come to *represent* action-guidingness per se. However, while I think this is implausible, fully assessing this alternative would require a longer departure into metasemantics, so I set it aside.

It seems to me that there are three other ways to try to preserve a KP version of CMP\*. First, there is the appeal to the cognitive permeation of some motivational state that was mentioned above. Second, one could deny the datum, arguing that CMP\* cannot directly motivate. Third, one could defend a metaphysics of perceptual experience which intrinsically incorporates non-conceptual conative states, such as what some theorists have claimed in the case of pain[22] [35,36]. These are all viable options. So, while BP can do potential explanatory work, this motivation is relatively weak.

*3.2. Grounding Moral Thought*

John Campbell has probably done more work than anyone in arguing that naive realism (the 'relational view', as he calls it) acquaints us with objects and properties in a way that allows for demonstrative reference. We saw above that one way he motivates this is in terms of what many find so unsatisfying about Putnam's brain-in-vat argument. But more directly and positively, Campbell argues that naive realist access to objects and properties is a prerequisite for the possibility of thought about objects and properties themselves (as opposed to thinking of their causal effects):

> Experience of the object has to explain how it is that we can grasp demonstratives referring to the object as referring to a categorical object, not merely a collection of potentialities. This means that, given any description of the phenomenal character of experiences of objects, we can ask whether experience, so described, would be capable of explaining our grasp of a demonstrative referring to the thing ... Merely having sensations could explain how it is that you have the conception of the object as a hypothesized cause of those sensations ... but it could not provide you with knowledge of the categorical thing itself ... that is exactly what happens when you rely on your experience of the object to interpret a demonstrative referring to that object. [11], p. 145.

Translating this into the language of Kantian vs. Berkeleyan Perception, Campbell is making the claim that demonstrative reference presupposes (because it is explained by) Berkeleyan Perception. This applies to demonstrative reference to objects as well as to properties. Kantian Perception, for Campbell, could provide us with propositional knowledge about the functional relations between objects and properties, but it can't provide knowledge of the objects and properties themselves. That is, if the traditional notion of *Objectual Knowledge* can be made sense of over and above knowledge of a cluster of propositions, Campbell thinks it requires Berkeleyan Perception of the object in question. The analogous knowledge in the case of properties is known as *Quiddistic Knowledge* [37].

This notion of Quiddistic Knowledge raises two questions for the moral perceptualist: (a) Is there special reason to think that we have Quiddistic Knowledge of (any of) the moral properties via perception? (b) Are there theoretical reasons to think that Quiddistic Knowledge is important for a broader metaethical theory?

Begin with (a). Aside from one issue—our epistemic access to normativity-as-such, which I discuss in the next section—it isn't clear exactly what Quiddistic Knowledge of moral properties would look like. On the view being considered here, non-Quiddistic Knowledge of some property $F$ consists in propositional knowledge of $F$'s structural location within a broader functional nexus of properties. Usually, the structural location of $F$ will be fixed by the variety of causes and effects that *Fness* has. This raises a bit of a puzzle in the normative case for two reasons. First, it is contentious whether normative properties have any causal powers [38–40]. Second, whether or not normative properties do have causal powers, surely a crucial feature of some normative property $N$'s role in an explanatory nexus has to do with $N$'s relations of *normative* support for and against actions, states-of-affairs, and other normative properties. Tentatively, then, let's focus on relations of normative support as those features of normative properties such that knowledge of them doesn't require Quiddistic Knowledge.

If we are looking for something more than knowledge of relations of normative support, what would it look like? I confess that I am not quite sure how to make sense of

this question. David Lewis, in his discussion and defense of Ramseyan Humility, considers whether qualia, pure qualitative properties, may be such that experiencing a qualitative property, in virtue of that experience, "knows just which property it is" [41], p. 217." Does experiencing, say, *goodness*, allow us to 'know just which property' *goodness* is? It *may* seem that experiencing *goodness* qua goodness involves understanding that it merits certain responses or actions. But notice that this is not Quiddistic Knowledge, but knowledge of relations of normative support.

It seems, then, that we have no clear reason to think that moral perception gives us Quiddistic Knowledge[23] [42–44]. Turning to (b)—is Quiddistic Knowledge of moral properties important for some aspect of broader metaethical theory? I think the answer to this question is largely negative. Quiddistic Knowledge doesn't seem relevant to one's ability to perform the right actions. If one has propositional knowledge, and this propositional knowledge is properly integrated with one's moral motivations, she has acted rightly[24]. But what about Campbell's argument, that—to put it in our current terminology—demonstrative reference to properties, which grounds thought about those properties, requires Quiddistic Knowledge? If Campbell's argument is true generally, then we need acquaintance with moral properties, either perceptually or through some other mechanism as a ground for the possibility of moral metasemantics. In contrast to this, the most popular metasemantic view for morality, or at least for moral realists, is conceptual role semantics (plus, perhaps, reference magnets), which does not require acquaintance with the properties in question[25] [45–48]. So it does not seem that those worried about reference to moral properties need to presuppose Quiddistic Knowledge.

*3.3. Epistemic Access to Normativity*

There remains one closely related motivation for Berkeleyan Property perception, which is raised by Logue herself:

> [I]t seems that veridical experience gives us something that trustworthy, reliable, and quickly delivered testimony doesn't. I propose that the 'something more' is something along the lines of the following: the *phenomenal character* of veridical experience gives its subject *insight into what things in one's environment are like independently of one's experiences of them.* [1], p. 227.

Logue goes on to argue that this can only be explained, for a given property and a given perceptual experience, if the property is a BP. We can state this thought more precisely as follows:

> *IKf > BPf.* For any perceptible property *F* in experience e, if e provides intrinsic knowledge of *F*, then *F* is a BP in e.

This principle is similar to, but weaker than, the notion of Quiddistic Knowledge. Quiddistic Knowledge purports to allow one, in principle, to track a property (or object, in the case of Objectual Knowledge) across modal space. It provides an understanding of a property such that one can individuate it from other, similar properties. "Intrinsic Knowledge" of a property may not provide enough information to grasp the property's full essence, or to individuate it from all similar such properties across modal space [1], pp. 228–229. But intrinsic knowledge is knowledge of at least some non-relational feature(s) of a property. For example, consider two ways of thinking about an experience of a specific shade of red, $red_{542}$. If experiencing $red_{542}$ gave us Quiddistic Knowledge, it would put us in a position to individuate $red_{542}$ from any other property, to know *which* property it is, independent of its relational properties[26] [49]. On the other hand, we may be in a position to have some intrinsic knowledge of a property like $red_{542}$ without having something as strong as Quiddistic Knowledge. For example, we may know, from visual experience, that $red_{542}$ is essentially a color property, or perhaps that it is only able, given its intrinsic nature, to play certain kinds of roles in the causal nexus[27]. According to IKf > BPf, if experience can provide us with knowledge of some property's intrinsic nature, and not merely its causal roles, our experience of that property must be Berkeleyan.

Turning to moral perception, we can now ask whether there is any intrinsic knowledge that experience of moral properties provides us. One might think that, if what I said above about moral properties and Quiddistic Knowledge is correct, the answer must be no. After all, if moral properties don't *have* quiddities, then what intrinsic features are even there to be grasped? But intrinsic features need not be quiddistic in the sense discussed in the previous section. Return to our toy example of $red_{542}$. Acquaintance with $red_{542}$ could provide knowledge of its intrinsic status as a color property. This is a feature of $red_{542}$ that is true just in virtue of itself. However, it's a second-order property that is shared by any number of other properties, and so is not the kind of individuating *knowledge-which* that would constitute Quiddistic Knowledge[28].

We have a natural candidate for a second-order property shared by the moral properties: Their status as *normatively authoritative* properties. First, let me say a bit about what I mean by *normatively authoritative* and why it's plausible that moral properties count as normatively authoritative. Then, I'll turn to the question of why the possibility of epistemic access to this feature will be important to many moral realists. This will provide the foundation for the most intriguing case for moral perception as Berkeleyan.

### 3.3.1. Normative Authority

Normative facts can come in different flavors. While there remains some plurality in the terminology used to divide these flavors up, I will follow one standard usage and distinguish between *formal* and *authoritative* normativity[29] [50–53]. Formal normativity is, in an important sense, 'cheap' and, in Howard & Laskowski's phrase, not inherently "deliberation-worthy"[30] [50]. In some sense, according to the rules of etiquette, one *ought* to place the salad fork on the outside of the dinner fork. But if one doesn't care about the rules of etiquette, and there is no further etiquette-independent reason to follow the rules of etiquette, the rules of etiquette can simply be ignored. On the other hand, authoritative normativity retains its normative force regardless of some agent's contingent desires or lack thereof. One cannot avoid criticism with respect to authoritatively normative oughts simply by stating that they don't care. Someone not caring about morality, for example, does not get them off the hook of moral requirements—if anything, it makes them even *worse*, morally speaking.

This characterization of the distinction between formal and authoritative normativity is merely gestural, and actually can't quite be right. Sometimes authoritative normativity is analyzed in terms of its "inescapability"[31] [54]; a domain is authoritative just in case an agent can't 'get out of' that domain's demands. Or perhaps a domain is authoritative just in case it is self-endorsing: One of its norms is that everyone must follow its norms. These analyses won't work, because they can be met by domains that are obviously merely formally normative[32]. Instead, authoritative normativity is increasingly taken to be unanalyzable, only explained via ostension[33] [55]. But what is important for present purposes is that authoritative normativity involves only those normative facts, domains, and properties that are inherently "deliberation-worthy".

### 3.3.2. Individuating the Authoritatively Normative

Proponents of morality as authoritatively normative generally motivate this idea by pumping intuitions about which (normatively authoritative) reasons agents have in a variety of cases[34] [55–57]. This method presupposes that we have some kind of grip on normative authority and which domains it plausibly applies to. Much less work has been done on the epistemology of normative authority. One place that a related issue comes up is in debates about alternative normative concepts[35] [58]. The problem of alternative normative concepts, as I'll use it here, concerns how we are to decide, between two (or more) coherent systems of normative properties, which system(s) we should be normatively guided by[36] [58–61]. A common response to this problem is to argue that some system of

normative concepts refers to the properties that are normatively authoritative, and these concepts are the properties that provide genuine authoritative normative guidance.

While these debates typically concern domains like *reasons* and *counter-reasons*, or *morality* and *shmorality*, structurally similar questions will arise with respect to individuating morality from a domain such as etiquette. However, it is already taken as a given that, if any domains are normatively authoritative in this robust sense, morality is one of those domains, and etiquette is clearly not. But this raises the question—on what epistemic grounds are we able to draw this distinction between domains?

### 3.3.3. Berkeleyan Perception and the Epistemology of Normative Authority

Moral epistemology—as with most normative epistemology—is largely concerned with justification, knowledge, and warrant for our first-order normative beliefs. At present, we are concerned with a second-order claim about a property that moral properties have:

> *Normative Authority* (NA)*:* Moral properties, or the moral principles that ground their instances, are *normatively authoritative* in the sense of being intrinsically and objectively deliberation-worthy.

As mentioned above, exactly how to precisely characterize normative authority is a matter of debate, and I don't have anything to add to that debate here. The reader is welcome to substitute in her favorite characterization for my approximate one.

Consider two ways one could deal with the epistemology of NA[37]. On one approach, the epistemology of first-order moral facts and the epistemology of NA are independently dealt with by pointing to distinct mechanisms. For example, perhaps moral perception provides access to first-order moral facts, but NA is a conceptual truth, and so known in virtue of understanding the meanings of the terms in NA. On a second approach, the epistemology of first-order moral facts and the epistemology of NA are treated as, if not unified, at least in some sense springing from the same epistemic source. Both approaches are perfectly legitimate, though it does seem that other things equal a more unified approach would be preferable. I can't provide a full defense of this claim here, but notice that our epistemic access to both types of moral facts—first-order and NA—are subject to serious epistemic worries, especially for the moral realist. If we had an integrated story of how we came to know both of these types of facts, then so-called epistemic 'queerness' concerns could be undermined in one fell swoop.

Let's finally return to what all of this background has to do with Berkeleyan Perception. Recall that, according to *IKf > BPf,* if perceptual experience of some property F provides knowledge of some intrinsic feature of F, then that experience of F is Berkeleyan. With that in mind, if some perceptual experience of some moral property M can provide knowledge that M is normatively authoritative, then we would have moral properties as BPs.

Perhaps this gives us reason to hope or wish that moral perception can be Berkeleyan, but does it provide evidence, even defeasibly, that moral perception *is* Berkeleyan? This is a difficult question to assess without considering alternative explanations of the epistemology of NA. However, if we have knowledge of NA, and Berkeleyan moral perception can explain it, this is certainly some reason to raise our credence in it, at least until we have alternative equally good or better explanations of the epistemology of NA. Furthermore, it's at least worth noting that when we consider the phenomenology of moral perception, descriptions of it often do appear to incorporate a sense of normative authority. See, respectively, the following quotes by Mandelbaum, FitzPatrick, and Bedke:

> This feeling of obligation appears as independent of preference, as many of the alternatives within our experience do not. Where neither alternative has this character, where our choices are wholly matters of preference or desire, the choice which we face does not appear as a moral choice. However, let either alternative appear not as a preference, but as an "objective" demand, and I feel myself to be confronted by a moral issue, by a categorical imperative, by an injunctive force which issues from one of the alternatives itself. [62], p. 50.

Your moral experience is not simply that of being required to do something by objective features of the circumstances you face given a certain set of standards to which you happen to be committed. A crucial part of the phenomenology is the powerful sense that the standards you are employing are themselves imposed upon you independently of your choices or contingent commitments or causal psychological shaping by your society. Not only are the wrong-making features of walking away from the child objective, along with their relation to a given set of standards, but their wrong-makingness itself seems to be objective, which is to say that the associated standards themselves have an objective status. The moral experience is that of being confronted by a moral demand that is backed up by categorically authoritative standards to which you are committed because they objectively merit that commitment—not because you have simply been raised to be so committed or made the choice to be. [63], p. 26.

If we start with normative concepts, it seems apt to characterize their normativity in terms of a special mode of presentation in cognition—that of inherent, authoritative guidance. Though there might be more to it than a distinctive phenomenology, the phenomenology seems really important. [64], p. 123.

These three theorists are each working within distinct theoretical frameworks of morality, yet their descriptions of the phenomenology of moral experience all have a ring of an experience which contains normative authority—or knowledge of it—as an important constituent. While it is true that among these three, only Mandelbaum is discussing moral perceptual experience, FitzPatrick and Bedke are, as far as I can tell, neutral about the nature of moral experiences, and so their descriptions are compatible with a perceptual understanding as well.

It appears, then, that we have one strong, albeit defeasible, reason to endorse an account of moral perception as Berkeleyan Perception. The phenomenology of moral perception suggests at least the appearance of access not just to moral properties, but to an important aspect of their intrinsic nature, NA. Furthermore, at least anti-skeptical realists about NA—which most moral realists are—need to provide an epistemological account of how we come to know that moral properties have normative authority. Endorsing the idea that moral perception is Berkeleyan provides such a story. Whether to ultimately accept the thesis that moral perception is Berkeleyan depends on other issues which, as noted above, can't be explored here. Among others, (a) Whether Berkeleyan Perception is ever possible; (b) The metaphysics of moral properties; (c) The metaphysics of normative authority (including whether it even exists); and (d) What alternative accounts of the epistemology of normative authority might be developed. Nonetheless, this is perhaps the most promising avenue for a new theoretical role that moral perception could play in a broader complete metaethical theory.

## 4. Two Worries from Moral Disagreement: Illusions and Cognitive Penetration

I have argued that a Berkeleyan conception of moral perception can provide an epistemology of normative authority. However, there are two potential problems here, each loosely related to the fact that there is widespread moral disagreement[38]. The first worry is connected to the possibility of illusory or hallucinatory moral experiences. The second worry is connected to cognitive penetration/permeation[39]. I address each in turn.

As with other perceptual experiences, moral perceptualists should grant that people can often be mistaken about the presence of moral properties. They can have illusory or hallucinatory experiences which are phenomenally indistinguishable from genuine perceptions of moral properties. These illusory perceptions will seem to the subject just as normatively authoritative as genuine perceptions. And it may seem even worse since, unlike many perceptual properties, the pervasiveness of moral disagreement suggests that these illusions will be widespread. Given that the subject cannot, by stipulation, differentiate

between genuine and illusory acquaintance with the normatively authoritative properties, how could Berkeleyan moral perception ground knowledge of normative authority?

This argument has quite a bit of intuitive force, but it should not cause any *new* concern for the moral perceptualist, since it is an argument that those who endorse Berkeleyan Perception of some property will independently reject. This is because, insofar as one is a Berkeleyan about some particular property *F* in experience, she ought to be an epistemological disjunctivist about *F*-perception. Epistemological disjunctivism is the view that the epistemic status of a given perceptual state can depend on factors independent of the subjectively accessible phenomenology. This means that whether some purported *F*-perception can be the basis for knowledge of *F* will depend in part on whether it is a genuine, Acquaintance-involving *F*-perception, or if it is merely an illusion[40] [65]. Disjunctivists of many stripes have independently defended this epistemic condition—indeed, it forms the basis of one of McDowell's arguments in favor of (metaphysical) disjunctivism about perceptual experience [66]. This leaves questions open about whether an illusory moral experience could provide *justification* for beliefs about normative authority, but that is as it should be.

The second worry concerns the existence of cognitive permeation. Cognitive permeation occurs when one's background cognitive states (beliefs, desires, etc.) influence and alter the contents of one's perceptual experience [67]. There is extensive evidence that cognitive permeation exists, though its pervasiveness is still a hotly disputed matter in cognitive science[41] [68,69]. Furthermore, the idea of moral beliefs cognitively permeating in the case of moral perception is often appealed to as a theory of how perceptual experience could contain moral properties, as well as explaining widespread perceptual moral disagreement[42] [70,71] However, if moral perception is the result of cognitive penetration, this suggests that, even when it is veridical, it is Kantian. After all, cognitive permeation suggests that the subject's background states contribute more to the phenomenology than the worldly property[43] [72].

This is, I take it, a strongly suggestive reason to favor a Kantian reading of moral perception. So it isn't just an objection; it's also a positive reason why moral perception might be Kantian. The Berkeleyan moral perceptualist may say two things in response. First, she could deny that cognitive permeation plays a central role in the perception of moral properties. This may seem implausible, but it's prima facie possible that one could perceive moral properties through a kind of perceptual learning, and models of perceptual learning differ on when cognitive permeation is necessary. A second response is more complicated. Recall that, although I've been speaking as though BP and KP are, for any given property in experience, a discrete fact, this isn't so. Rather, perception of properties exists on a continuum depending on how much and in what ways the subject contributes to the phenomenology of the property experience. This gives space for a view according to which cognitive permeation influences the detection of moral properties, but the perceptual experience still involves acquaintance with the normative authority of the property in question. As it stands, such a response may appear somewhat ad hoc. However, I think its ultimate plausibility depends on several complicated issues about the epistemology and metaphysics of cognitive permeation which I can't go into here. Suffice it to say, this looks like the right place for the Kantian to press on the view tentatively defended above[44,45] [71].

## 5. Taking Stock

Proponents of moral perception have spent the bulk of their time in recent years defending the view from some of the most pressing objections raised against it. Some recent work in the area has attempted to show, in more detail, some of the epistemic and metaphysical benefits of moral perception[46] [73,74]. These are both worthwhile kinds of projects. But proponents of moral perception have been less engaged in how their view(s) square with background work in the philosophy of mind and philosophy of perception. In this paper, I have taken one small step toward remedying this by considering the relationship between moral perceptual experience and a foundational debate in philosophy of

perception on the metaphysics of perceptual experience. In particular, I have explored a few reasons why the moral perceptualist may favor an account of moral perceptual experience such that moral properties are constituents in experience, rather than merely represented in experience. My results were unfortunately (but perhaps realistically) tentative.

I first considered whether conceiving of moral perception as Berkeleyan provides a better theory of non-reflective moral responses to perceived situations. I concluded that it can provide *a* nice explanation of non-reflective moral actions, but that there are certainly other equally plausible alternative explanations of the same phenomenon. Next, I considered whether Berkeleyan moral perception could provide superior alternatives for moral metasemantics as compared to Kantian moral perception. I argued that there is no special reason to think so, unless a very general argument in favor of naive realism from Campbell (2002) works, in which case the moral component of moral perception wouldn't be playing any special role. Finally, I considered the idea that moral perception as Berkeleyan could provide an Acquaintance-based epistemology of normative authority. While not all metaethicists take normative authority (in the technical sense discussed here) seriously, I argued that insofar as one does want an account of the epistemology of normative authority, a Berkeleyan account of moral perception is in a good position to provide such an account, whereas a Kantian account is not. This is far from proving that moral perception must be Berkeleyan, even for the realist about normative authority. However, insofar as there are no other satisfying accounts on offer, this is a potential theoretical gain that is far from trivial.

In short, then, the existence of moral perception is only one important question in this area; depending on one's other commitments, the nature of moral perception will also have important implications for the theoretical roles that moral perception can and cannot play within one's broader metaethical theory.

**Funding:** This research received no external funding.

**Institutional Review Board Statement:** Not applicable.

**Informed Consent Statement:** Not applicable.

**Data Availability Statement:** Not applicable.

**Conflicts of Interest:** The author declares no conflict of interest.

**Notes**

1. See https://survey2020.philpeople.org/survey/results/4894, accessed on 22 May 2023.
2. This is a simplified statement of Logue's view—for the details, see Section 1.3.
3. Cowan (2015, 167) was the first to point this out. See also, McBrayer (2010), Audi (2013), Ch.2, Werner (2016), Väyrynen (2018), Matey (forthcoming) [3–7].
4. Strictly speaking, this is a matter of a degree, rather than a hard-and-fast distinction. See Section 1.3.
5. And in fact, it isn't even clear that full-blown representation *is* incompatible with Berkeleyan Property perception. See Section 1.3, especially footnote 20.
6. See Fish (2021 Ch.1) [8].
7. For ease of reading, I'll leave this qualification about context implicit in the remainder of what follows.
8. Fish (2021), Ch.3 [8].
9. Representationalism, or intentionalism, comes in many different forms and strengths, which I will not be delving into here. For details, see Lycan (2019), Section 2 [9].
10. I have in mind here primarily the detailed articulation and development of various forms of disjunctivism. See the variety of perspectives in Byrne & Logue (2009) [12].
11. Russell (1912). For a summary, see Hatfield (2021), Section 2.2.2 [13,14].
12. See, for example, Campbell (2002), Fish (2009, 75), and Langsam (2017) [15,16].
13. One difference between BK-perception and edenic perception is that Chalmers appears to allow for the possibility that edenic perception is representational. While Logue allows for such a view (see Logue 2012, 225–226), she thinks such a view will fail to have the epistemological advantages that a naive realist conception of BK-perception will have.

14    Werner (2020), p.6. Bergqvist & Cowan (2018) call this "canonical moral perception" [18].

15    For an overview of the most serious objections, see Werner (2020), Section 4. See also McBrayer (2010), Reiland (2021) [19,20].

16    And, in fact, some arguments for moral perception appeal to the anti-representationalist tradition of Gibsonian ecological perception. See Hamilton (2020), Van Grunsven (2022), Wisnewski (2015, 2019) [21–24].

17    For the former, see Murdoch (1970), Blum (1994), Vance & Werner (2022). For the latter, see Cowan (2014, 2015), Werner (2016, 2018) [4,25–30].

18    Notice that this is distinct from an argument to the effect that appealing to a background desire would be generally a more complex explanation of action, and so this is not a general argument against Humeanism about belief/desire explanations. This is because the complexity here is not about how many states are involved in any given action explanation, but about the cognitive mechanisms—and their realisticness—in a given subset of action explanations.

19    Two anonymous reviewers worry that this kind of story can't explain why illusory/hallucinatory (purported) perceptions of normative properties would be motivating in the same way, since they by definition do not provide *Acquaintance*. I return to this important issue in Section 4 since an analogous concern arises with respect to knowledge of normative authority.

20    For an overview, see Björnsson et al. (2015) [34].

21    Thanks to an anonymous referee for pressing me on this.

22    See Barlassina & Hayward (2019), Jacobson (2021) [35,36].

23    In fact, there might be a positive reason to think that we obviously *don't* have Quiddistic Knowledge of moral properties, because if we did, questions about the metaphysics of moral properties would have been settled long ago merely by introspection. If a version of this argument succeeds, it would indirectly tell in favor of moral properties in perception as KPs. I'm not sure that the argument can work, however. After all, those who endorse Quiddistic Knowledge for, say, qualia, do not take their views to be refuted by the fact that there remains a dispute about the metaphysics of qualia. See Majeed (2017) and Liu (2020, forthcoming) for discussion of some of these issues [42–44].

24    There may be further conditions on acting rightly or on acting with moral worth, but I know of no account of such things that requires anything resembling Quiddistic Knowledge.

25    The most detailed defenses of conceptual role semantics for normative terms are Wedgwood (2007) and Dunaway (2020). See also Enoch (2011, 7.6), and Suikkanen (2017) [45–48].

26    This is not to say that the knowledge needs to be even in principle stateable. Knowledge-which may be ineffable. See Dasgupta (2015) [49].

27    Compare understanding the definition of a word vs. understanding that some particular word is a noun, and so only suited to play certain grammatical roles.

28    This is *not* to say that Kantian perception could not provide any knowledge of second-order properties, or that Kantian perception could not (re)present second-order properties. Rather, there is a certain kind of second-order property that KP cannot provide us, according to IKf > BPf, and that is knowledge of something like the intrinsic nature of some property—a sort of partial Quiddistic Knowledge. Of course, one could deny IKf > BPf, but I am taking it for granted in what follows. I thank an anonymous referee for pressing me to think more carefully about this issue.

29    McPherson (2011, 2018), Wodak (2019), Howard & Laskowski (forthcoming) [50–53].

30    Howard & Laskowski (forthcoming), 8. It should be noted that Howard & Laskowski are ultimately skeptical of authoritative normativity [50].

31    See Paakkunainen (2018), Section 6 [54].

32    The Inescapable Game has three rules. 1. Everyone must play the game, and 2. Everyone must harshly criticize anyone who is losing the game, and 3. The only ways to lose the game are (a) to claim that you aren't playing it or (b) to fail to criticize anyone who is losing the game. I refuse to play the Inescapable Game. But its reasons nonetheless apply to me. Claiming that I don't care about the game or its rules only results in further obligations that I will be harshly criticized.

33    One core source of this thought, though not expressed in these terms, is Parfit (2011), Volume 2, Part Six [55].

34    For examples of this strategy, see Parfit (2011), Part One, Shafer-Landau (2009), Luco (2016) [56,57].

35    Eklund (2017) is the most detailed exploration of these issues [58].

36    This presupposes that we can ask this question coherently, which is far from trivial, as Eklund (2017) argues. I will bracket this issue here. See Leary (2020), McPherson (2020), and Werner (2022) for discussion and responses [58–61].

37    It's worth flagging that similar such questions will plausibly arise with respect to some other metaethical claims, such as the supervenience of the moral on the natural.

38    I thank an anonymous referee for pressing me to address the first issue, and a different anonymous referee for pressing me to address the second issue.

39    Both terms are used for the same phenomenon. I will use "cognitive permeation" in what follows.

40    For discussion, see Soteriou (2020), Section 2.4 [65].

41    See Vetter and Newen (2014), Newen and Vetter (2017) [68,69].

42    Werner (2017), Elliott (forthcoming) [70,71].

43    Cavedon-Taylor (2018) has suggested that cognitive permeation supports representationalism generally, but even if the argument does not work at complete generality, it could provide a test for whether a given perception of a property was Kantian [72].

44    One issue here has to do with whether cognitive permeation genuinely is incompatible with the Berkeleyan perception of some property. It's known, for example, that gray bananas appear more yellow than they in fact are (Macpherson 2012). Suppose I see a banana in a dark room, and it appears somewhat yellow to me. Furthermore, suppose that the banana is in fact yellow, and the cognitive influence is merely correcting for error predictably produced by the darkness of the room. Does this undermine my perception of the *yellowness* as Berkelyan? It seems initially like the answer is yes; however, my perception of the banana as yellow could be based on prior Berkeleyan perceptions of *yellowness* in bananas, so the ultimate explanation of my experience is given in terms of the worldly facts and not the subject [67].

45    I think what this brings out is that the distinction between Kantian and Berkeleyan property perception is more complicated than Logue suggests, because it isn't clear what range of facts count as part of the explanation of the phenomenology in the relevant sense.

46    Samuel (2021), Werner (forthcoming) [73,74].

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
