# Peer review of "How Naive Is Contentful Moral Perception?"

_philosophies, doi:10.3390/philosophies8030049_

Round 1

Reviewer 1 Report

Summary:

This manuscript is on questions about the nature of moral perception. It has three main parts. The first one outlines the lay of the land in philosophy of perception. It first explains the main views, representationalism and naïve realism, and then Heather Logue’s version of moderate naïve realism which the manuscript thereafter will assume. This view draws a distinction between Kantian and Berkeleyan properties, where the difference is between how much our own features as perceiving subjects contributes to the phenomenology of perceiving the properties when they constitute our perceptions. For the Kantian properties, our own features contribute significantly whereas for the Berkeleyan properties they do not. This has consequences for whether we can be said to have direct awareness of the properties.

The key question of the manuscript is then introduced in section 2. The question is, if we assume that we have moral perceptions and that moral properties constitute these experiences (in the manner of naïve realism) should we think of the moral properties in these experiences more as Kantian or Berkeleyan properties?

The section 3, which is the main bulk of the manuscript, then considers what reasons there might be for thinking of the moral properties as Berkeleyan properties, as properties that constitute our moral perceptions in a manner in which our own features as the subjects of these experiences do not make a significant contribution. The manuscript considers three considerations that could be said to favour this view. It could be argued that the moral properties, as Berkeleyan properties, could (i) be used to explain how moral perceptions motivate us or that moral perceptions of Berkeleyan properties could help to explain either  (ii) the Quiddistic knowledge we have of the moral properties in moral perceptions (and thus explain how our moral terms come to refer to the moral properties) or (iii) slightly more weakly the moral knowledge we have of these properties as Normatively Authoritative.

The conclusions in these latter sections are somewhat tentative. It seems to be concluded that (i) offers only weak support for the moral properties are Berkeleyan view, (ii) offers no support at all, and (iii) offers perhaps the most compelling argument for the view even if here too there are various complications.

Evaluation

Overall, I believe that this is an excellent manuscript and so I wholeheartedly recommend its publication in Philosophies. The manuscript masters large areas of literature not only about the debates on moral perception but also concerning metaethics and philosophy of perception more broadly. It is clearly written and structured, it is knowledgeable, and most importantly it asks a wholly new question in a way that makes a genuine contribution to thinking about moral properties. As the author notes, moral perception has been discussed in terms of moral representationalism and so formulating and developing a naïve realist alternative is something new and very interesting. And, the fact that the conclusions about the arguments for the view are quite tentative does not make the manuscript any less important as the manuscript really prepares ground for future debates more than vindicates a specific position in the debate. But, seen in this way, I do think that the manuscript is very successful.

I do also have just two related comments I would wish the author to address before the submission of the final draft (and a couple of types/formatting issues).

Page 8, the middle paragraph, the discussion on BPs and moral motivation. The author writes: ‘On the other hand, Berkeleyan Perception of moral properties would not just tell a perceiver that some moral properties are present, but may also provide access to their normative force. Acquaintance (in the sense of Berkeleyan Perception provides) with a moral property would  pack in this normative component in a way that makes spontaneous non-deliberative motivational response rational.”

Just one concern here: Note that cases of hallucinations/illusions/other perceptual errors has always been an issue for naïve versions of realism. But, here the issue is slightly different. It is well-known that false moral perceptions (and surely there must be some according to the discussed view) are just as motivating – the spontaneous non-deliberative motivational responses are just as rational towards them. If this is right, then it is harder to see how the motivation or its rationality could have a source in the perceived properties that constitute the perceptions. Ralph Wedgwood’s ‘Metaethicists’ Mistake’ is a good paper on this.

Page 14, last full paragraph. I have a similar concern about the idea that perception of Berkeleyan properties is needed for an epistemic access to the intrinsic nature of these properties and especially their normative authority. Again, it seems that, given the amount of fundamental moral disagreement, there are people who misperceive moral properties – they fail to perceive the moral properties that exist and perceive moral properties where they are not instantiated. But, it doesn’t seem that the misperceiving subjects think that moral properties are any less categorical. This again makes it difficult to see how the knowledge of categoricity could come from the properties themselves.

Thus, it would be nice to see the author to address these related concerns in the final manuscript.

Otherwise, here a few typos:

Page 7, 290: there is a double full stop.

Page 8, 318: something wrong with the grammar.

Page 14: the long quotes from Fitzpatrick and Bedke should be formatted as block-quotes like the Mandelbaum quote before them.
